

**Variations in $N_{cn}$ and $N_{ccn}$ over China marginal seas related to marine traffic**
**emissions, new particle formation and aerosol aging**
Yang Gao[1,2#*], Deqiang Zhang[1#], Juntao Wang[1], Huiwang Gao[1,2] and Xiaohong Yao[1,2*]
[1]Frontiers Science Center for Deep Ocean Multispheres and Earth System, and Key Laboratory
of Marine Environment and Ecology, Ministry of Education, Ocean University of China,
Qingdao, 266100, China
[2]Laboratory for Marine Ecology and Environmental Science, Qingdao National Laboratory for
Marine Science and Technology, Qingdao, 266237, China

9                    [#]Authors contribute equally to this study

[*]Correspondence to yanggao@ouc.edu.cn; xhyao@ouc.edu.cn





**Abstract**

In this study, a cruise campaign was conducted over China marginal seas to measure
concentrations of condensation nuclei ($N_{cn}$), cloud condensation nuclei ($N_{ccn}$) and other
pollutants during DOY 110 to DOY 135 of 2018. With exhaustedly excluded self-ship
emission signals, the mean values of $N_{ccn}$ during the cruise campaign slightly increased
from $3.2 \pm 1.1 \times 10^3$ cm$^{-3}$ (mean ± standard) at supersaturation (SS) of 0.2% to $3.9 \pm$
$1.4 \times 10^3$ cm$^{-3}$ at SS of 1.0%, and the mean value for $N_{cn}$ was $8.1 \pm 4.4 \times 10^3$ cm$^{-3}$. Data
analysis showed that marine traffic emissions apparently yielded a large contribution to
the increase of $N_{cn}$ in daytime, especially in marine atmospheres over their heavily
travelled sea zones; however, the fresh sources had no clear contribution to the increase
of $N_{ccn}$. This finding was supported by the quantitative relations between $N_{cn}$ and $N_{ccn}$
at SS=0.2-1.0% against mixing ratios of $SO_2$ in self-ship emission plumes, i.e., 1 ppb
increase in $SO_2$ corresponds to $1.4 \times 10^4$ cm$^{-3}$ increase in $N_{cn}$, but only 30-170 cm$^{-3}$
increase in $N_{ccn}$ possibly because of abundant organics in the aerosols. The smooth
growth of marine traffic derived particles can be observed, reflecting aerosol aging. The
estimated hygroscopicity parameter ($\kappa$) values were generally as high as 0.46-0.55
under the dominant onshore winds, suggesting inorganic ammonium aerosols likely
acting as the major contributor to $N_{ccn}$ through aerosol aging processes largely
decomposed organics. Moreover, the influences of the transported new particles from
the continent on $N_{cn}$ and $N_{ccn}$ in the marine atmosphere were also investigated.

**Key words:** $N_{cn}$; $N_{ccn}$; marine traffic emissions; hygroscopicity parameter; $SO_2$



## 1. Introduction

Oceans occupy approximately 2/3 of the Earth's surface and water evaporation from oceans acts as the major source of moisture in the atmosphere. Aerosol-cloud interactions in marine atmospheres, covering from tropics to polar regions, thereby attract great attentions in the past few decades due to their impact on the climate change (Brooks and Thornton, 2018; Huebert et al., 2003; Quinn and Bates, 2011; Rosenfeld et al., 2019; Wang et al., 2014; Yu and Luo, 2009). However, large uncertainties still exist in various marine atmospheres, e.g., the sources of aerosols, concentrations of bulk cloud condensation nuclei (CCN) and aerosol CCN activation under various of supersaturation, etc. (Clarke et al., 2006; Decesari et al., 2011; Quinn and Bates, 2011; Rosenfeld et al., 2019; Saliba et al., 2019). The uncertainties are mainly determined by limited observations in marine atmospheres, although a few additional observations of number concentrations of aerosol ($N_{cn}$) and CCN ($N_{ccn}$) were recently reported in different marine atmospheres, e.g., over Mediterranean, Sea of Japan, Bay of Bengal, coast of California and the Northwest Pacific Ocean etc. (Bougiatioti et al., 2009; Ramana and Devi, 2016; Ruehl et al., 2009; Wang et al., 2019; Yamashita et al., 2011).

Including sea-spray aerosols and secondarily formed aerosols from sea-derived gaseous precursors (Blot et al., 2013; Clarke et al., 2006; Fossum et al., 2018; O'Dowd et al., 1997; Quinn and Bates, 2011), marine traffics also emit a large amount of aerosols and reactive gases (Chen et al., 2017a). These pollutants may also directly or indirectly contribute to CCN therein, to some extent (Langley et al., 2010). In addition, the long-range transport of continental aerosols widely reportedly acted as an important source of CCN in marine atmospheres (Charlson et al., 1987; Fu et al., 2017; Huebert et al., 2003; Royalty et al., 2017; Sato and Suzuki, 2019; Wang et al., 2019). The continent-derived aerosol particles observed in marine atmospheres usually mix with different sources such as biomass burning, dust and anthropogenic emissions (Feng et al., 2017; Guo et al., 2014; Guo et al., 2016; Lin et al., 2015). An appreciable fraction of organics



reportedly exists in marine aerosols and continental aerosols upwind of oceans (Ding
et al., 2019; Feng et al., 2012; Feng et al., 2016; O'Dowd et al., 2004; Quinn et al., 2015;
Song et al., 2018). However, ammonium sulfate aerosols have been frequently reported
to dominantly contribute to CCN-related aerosols in many marine atmospheres and lead
to hygroscopicity parameter ($\kappa$) larger than 0.5 (Cai et al., 2017; Fu et al., 2017;
Mochida et al., 2010; Phillips et al., 2018; Royalty et al., 2017). A question is
automatically raised, i.e., where do particulate organics go in the marine aerosols
enriched in ammonium sulfate? Anthropogenic emission such as $SO_2$, $NO_x$ in general
increase since 1980s, until recently started to decrease, i.e., $SO_2$ start to decrease from
2006 (Li et al., 2017) whereas NOx started to decrease since 2011 (Li et al., 2017; Liu
et al., 2016). Together with the influence of the Asian Monsson, the marginal seas of
China are, therefore, inevitably affected by the outflow of continent al aerosols (Feng
et al., 2017; Guo et al., 2016). Observations of $N_{cn}$ and $N_{ccn}$ in marine atmospheres over
China marginal seas helps to resolve the data scarcity, understand their sources and
dynamic changes and better service the study of their potential climate impacts.

In this study, cruise campaigns were conducted to measure the $N_{ccn}$, $N_{cn}$, particle
number size distributions, gaseous pollutants and aerosol composition of water-soluble
ionic species over the marginal seas from 20 April 2018 (day of year (DOY) 110) to 15
May 2018 (DOY 135), traveling from the East China sea to the South China sea, and
returning to the Yellow sea. Spatiotemporal variations in $N_{cn}$, $N_{ccn}$ and CCN activities
of aerosol particles were studied. The *Kappa* values of aerosol particles from DOY 110
to DOY 118 over the marine were calculated and analyzed. Finally, we tried to establish
the correlations of $N_{cn}$ and $N_{ccn}$ with mixing ratios of $SO_2$ in self-ship plumes and
ambient marine air. The correlation equations are valuable for a rough estimation of $N_{cn}$
and $N_{ccn}$ from $SO_2$ when their direct observations are not available.

**2. Experimental design**
*2.1 Instruments and data sources*





A cruise campaign was conducted across China marginal seas, including the East China
sea, the South China sea and the Yellow sea from Day of Year (DOY) 110 to DOY 135
of 2018 (Fig. 1a,b). A suite of instruments including a Fast Mobility Particle Sizer
(FMPS, TSI Model 3091), CCN counter (CCNC, DMT Model 100), Condensation
Particle Counter (CPC, TSI Model 3775), gas analyzers, Ambient Ion Monitor-Ion
chromatography (AIM-IC) etc., were onboard a commercial cargo ship *Anqiang 87* for
measurements. The FMPS were used to measure particle number size distributions with
mobility diameters from 5.6 nm to 560 nm in 32 channels at 1-second temporal
resolution with an inlet flow of 10 L min$^{-1}$. The CPC were used to report the total
number concentrations of particles in the range of 3-3000 nm ($N_{cn}$) in 2-second time
resolution with an inlet flow of 1.5 L min$^{-1}$. The $N_{cn}$ was then used to calibrate the
particle number size distributions simultaneously measured by the FMPS, on basis of
the procedure proposed by Zimmerman et al. (2015). Due to the severe oceanic
condition and humid weather conditions, the FMPS and CPC were out of service after
DOY 118 and DOY 122, respectively. Prior to the campaign, the CCNC was calibrated
with ammonium sulfate particles based on the standard procedure detailed at Rose et al.
(2008). The total flow rate of CCNC was 0.45 L min$^{-1}$, with a ratio of sample to sheath
at 1/10, and five super saturations (SS) conditions were selected including 0.2 %, 0.4 %,
0.6 %, 0.8 %, and 1.0 %. More detailed information about the collection of CCN can
be found in Wang et al. (2019).

During the measurement, ambient particles were first sampled through a conductive
tube (TSI, US) and a diffusion dryer filled with silica gel (TSI, US), and then splitted
into different instruments with a splitter. All instruments were placed in an air-
conditioned container on the deck of ship, with inlet height of approximately 6 m above
the sea level. Regarding the gas analyzers, the ambient O$_3$ (Model 49i, Thermo
Environmental Instrument Inc., USA C-series), SO$_2$ (Model 43i, Thermo
Environmental Instrument Inc., USA C-series), and NO$_x$ (Model 42i, Thermo
Environmental Instrument Inc., USA C-series) were measured in mixing ratios with
temporal resolution of one-minute. The CCNC and gas analyzers were operated





properly throughout the entire campaign. The same was true for Ambient Ion Monitor-
Ion chromatography (AIM-IC), which was used to measure water-soluble ionic species
in ambient particles less than 2.5 µm.

In this study, the Hybrid Single-Particle Lagrangian Integrated Trajectory (HYSPLIT)
model from the NOAA Air Resources Laboratory was used to track the particle sources.
The input of HYSPLIT such as wind speed and wind direction was from European
Centre for Medium-Range Weather Forecasts (ECMWF). Meanwhile, the data of fire
spots was available at the Fire Information for Resource Management System
(FIRMS;http://firefly.geog.umd.edu/firemap).

The hygroscopicity parameter (κ) was calculated according to the method proposed by
Petters and Kreidenweis (2007).
$$\kappa = \frac{4A^3}{27D_d^3 \ln^2 S_C} \ , \ A = \frac{4\sigma_{s/a} \ M_w}{RT\rho_w}$$
where $D_d$ is the dry diameter, $S_C$ is the super saturation, , $M_w$ is the molecular weight
of water, $\sigma_{s/a}$, as a constant of 0.072 J m$^{-2}$, represent the surface tension over the
interface of the solution and air, R is the universal gas constant, T is the ambient
temperature and $\rho_w$ is the water density. The FMPS has a low size resolution,
particularly at the size greater than 90 nm, which doesn't allow accurately calculating
*Kappa* values at SS=0.2%. At SS=0.6% and 0.8%, the *Kappa* value was not calculated
considering the complication in the explanation of the value, possibly reflecting the
combined effects of particle size, mixing state and chemical composition.

*2.2 Separating ambient signals of $N_{cn}$ and $N_{ccn}$ from self-ship emission signals*
The data measured during the cruise campaign were frequently interfered by self-ship
emission signals. Ambient and self-ship emission signals of $N_{cn}$ and $N_{ccn}$ over the
marginal seas were first distinguished from each other and studied separately. The data
measured at 18:00-24:00 on DOY 115 were used to illustrate the separation, with the
size distribution of particle number concentration during DOY 110-118 shown in Fig.



S1-S9 in the supporting information. At 18:00-21:11 LT (Local Time), the low $N_{cn}$ of
$5.8\pm0.4\times10^3$ cm$^{-3}$ were observed. The accumulation mode dominated in particle
number concentration with the median mobility mode diameter at 105±4 nm (Fig. 2a).
Afterwards, the $N_{cn}$ rapidly increased by over one order of magnitude (Fig. 2b). The
dominant particle number concentration mode changed from accumulation mode to
Aitken mode, with the median mobility diameter of Aitken mode stabilized at 47±4 nm
in approximately 90% of the time. The rapid increase in $N_{cn}$ and the change in mode
size indicated the signal of ship emission itself. The self-ship emission interference after
21:11 was also supported by additional evidences, e.g., a large decrease in activation
ratio (AR), defined as the quotient of $N_{ccn}$ and $N_{cn}$, from >0.5 to <0.2 at SS=0.4% (Fig.
2c) due to large increase of $N_{cn}$ but much smaller magnitude enhancement of $N_{ccn}$ (Fib.
2b), a rapid increase of NO$_x$ from <10 ppb to 192±99 ppb, NO/NO$_2$ from <0.1 to
0.7±0.3, as well as SO$_2$ from <2 ppb to 6.2±2.4 ppb. The large changes were expected
because the ship smoke stack was approximately only 10 meters away from these
detectors. Thus, based upon the feature described above certain criteria were designed
in this study to identify self-ship emission signals so as to separate from ambient signals,
i.e., a net increase in $N_{cn}$ beyond $5\times10^4$ cm$^{-3}$ in five minutes, the median mobility mode
diameter around 50 nm, NO$_2$>30 ppb and NO/NO$_2$>0.5. Please note that there was a
short period lasting a few minutes during the transition of signals dominant by either
ambient environment or self-ship emission. which was excluded from the following
analysis.

**3. Results and discussion**
*3.1 Spatiotemporal variations in ambient $N_{cn}$ during the cruise period*
Fig. 3 shows a time series of minutely averaged distributions of $N_{cn}$, $N_{ccn}$ and AR at SS
of 0.4% and 1.0% from DOY 110 to DOY 135 2018, when self-ship emission signals
had been exhaustedly removed.

When spatiotemporal variations in $N_{cn}$ were examined during the first half cruise period
(Fig. 3a), the $N_{cn}$ spanned a broad range of $0.2-4.5\times10^4$ cm$^{-3}$ with the average value





of $8.1 \pm 4.4 \times 10^3$ cm$^{-3}$. Specifically, the $N_{cn}$ were only $6.5 \pm 0.8 \times 10^3$ cm$^{-3}$ at 00:00-
06:00 LT on DOY110 when the ship anchored at the Yangtze River estuary near
Shanghai (Fig. 1). The low $N_{cn}$ were comparable to the mean value of $N_{cn}$ ($5.4 \times 10^3$ cm$^{-}$
$^3$) in marine-air cases during January-December 2010 in Shanghai reported by Leng et
al. (2013). The $N_{cn}$ largely increased to $1.9 \pm 0.7 \times 10^4$ cm$^{-3}$ at 08:00-21:00 LT on
DOY110 when the ship cruised across the Yangtze River estuary. The measured
particles in number concentration were dominantly distributed at Aitken mode on that
day while the median Aitken mode diameter shifted from 49±5 nm at 00:00-06:00 to
40±5 nm at 08:00-21:00 (Fig. S1). The Yangtze River estuary contains several world-
class ports and is heavily travelled by marine traffics in daytime (Chen et al., 2017).
Since the onshore wind dominated on that day (not shown), the increase in $N_{cn}$ and the
decrease in median Aitken mode diameter at 08:00-21:00 LT possibly reflected the
increased contribution from marine traffic emissions. Marine traffics visibly decreased
when the ship left the Yangtze River estuary toward the south. The $N_{cn}$ were then
significantly decreased, i.e., $9.5 \pm 4.4 \times 10^3$ cm$^{-3}$ in the marine atmosphere over the sea
zone in Zhejiang Province (at 07:00 LT on DOY111 - 17:00 LT on DOY 114), with
P<0.01. The $N_{cn}$ further decreased to the lower values of $5.8 \pm 1.7 \times 10^3$ cm$^{-3}$ in the
marine atmosphere over the sea zone in Fujian Province (at 18:00 LT on DOY114 -
14:00 LT on DOY 117). All these values were, however, 1-2 orders of magnitude larger
than the background values in remote clear marine atmospheres, e.g., <300 particle cm$^{-}$
$^3$ without the influence of industrial activities in the western Pacific and the tropical
Pacific (Ueda et al., 2016) and those reported by Quinn and Bates (2011) and Saliba et
al. (2019), indicating overwhelming contributions from non-sea-spray aerosols
including marine traffic emissions, the long-range continental transport, newly formed
particles in marine atmospheres, etc. As reported, atmsopheric particles over China
marginal seas can be further transported to the remote northwest Pacific Ocean (NWPO)
in spring under westerly winds, e.g., $N_{cn}$ observed over the NWPO in March-April 2014
were as high as 2.8±1.0 $\times 10^3$ cm$^{-3}$ and approxiately half of that over China marginal
seas observed in March 2014 (Wang et al., 2019).

The mean value of $N_{cn}$ ($8.1 \pm 4.4 \times 10^3$) observed in this study was close to that of 7.6
$\pm 4.0 \times 10^3$ cm$^{-3}$ (the number concentrations of particles larger than 10 nm) over the
eastern part of the Yellow sea in spring 2017 reported by Park et al. (2018). They





attributed the high number concentrations of particles within nucleation and Aitken
modes to the long-range transport of air pollutants over eastern China under the
influence of westerly winds. Consistently, larger values of $N_{cn}$ were frequently observed
in the continental atmospheres upwind of the Yellow sea, e.g., the mean values of $1.8 \pm$
$1.4 \times 10^4$ cm$^{-3}$ in May 2013 in Qingdao, a coastal city in proximity to the Yellow Sea (Li
et al., 2015), $3.18 \times 10^4$ cm$^{-3}$ in February-August 2014 in Beijing (Dal Maso et al., 2016),
and $1.0 \times 10^4$ cm$^{-3}$ in continental-air cases during January-December 2010 in Shanghai
(Leng et al., 2013).

*3.2 Spatiotemporal variations in ambient $N_{ccn}$ during the cruise period*
$N_{ccn}$ data were generally available during the entire campaign (Fig. 3b). The mean
values of $N_{ccn}$ over China marginal seas during the DOY 110 to DOY 135, 2018 were
from $3.2 \pm 1.1 \times 10^3$ cm$^{-3}$ to $3.9 \pm 1.4 \times 10^3$ cm$^{-3}$ under SS ranging from 0.2% to
1.0%(Table 1), two to four times larger than the $N_{ccn}$ at the same SS over the NWPO in
March-April 2014 (Wang et al., 2019), and much higher, i.e., 1-2 orders of magnitude,
than the pristine marine background values (Quinn and Bates, 2011). As was discussed
in the previous section, the mean $N_{cn}$ in this study ($8.1 \pm 4.4 \times 10^3$ cm$^{-3}$) was comparable
to that of $N_{cn}$ ($7.6 \pm 4.0 \times 10^3$ cm$^{-3}$) over the Yellow Sea in spring 2017 in Park et al.
(2018); however, the comparison of mean $N_{ccn}$ reveals that mean value ($3.6 \pm 1.2 \times 10^3$
cm$^{-3}$) at SS of 0.6% in this study was approximately 25% smaller than that ($4.8 \times 10^3$
cm$^{-3}$ at similar SS of 0.65%) in Park et al. (2018), likely a result of long range transport,
considering the relatively distant (i.e., 500-600 km) observations away from the land
depicted in Fig. 1 of Park et al., 2018, and the subsequently higher extent of aerosol
aging. $N_{ccn}$ under SS of 0.2% in this study ($3.2 \pm 1.1 \times 10^3$) is comparable to that ($3.1 \pm 1.9$
$\times 10^3$) by Li et al. (2015) in the continental atmosphere of Qingdao in May 2013,
however, the increment of $N_{ccn}$ with the increase of SS was much weaker in our study,
resulting in on average of 36% smaller in $N_{ccn}$ under SS of 0.4% to 1.0% compared to
Li et al. (2015). Consistently, the sensitivity differences of $N_{ccn}$ to SS between relatively
clean (i.e., $N_{cn}$ ($8.1 \pm 4.4 \times 10^3$) in this study) and polluted (with Ncn of $1.8 \pm 1.4 \times 10^4$
cm$^{-3}$) environment in Li et al. (2015) is also reported by Nair et al. (2019), who found



little sensitivity of $N_{ccn}$ to changes in SS over the equatorial Indian Ocean (< 6 °N) with
relative clean air, and much larger enhancement of $N_{ccn}$ with the increase of SS in
polluted marine atmospheres (> 6 °N).

In addition, $N_{ccn}$ at SS from 0.1% to 1.0% during the period with high $NH_4^+$ (17:00 LT
on DOY 114 to 10:00 LT on DOY 120) is statistically significant higher (P<0.01) in
comparison to the poor $NH_4^+$ period (11:00 LT on DOY 120 to 7:00LT on DOY 136;
Fig. 3b). More specifically, a large increase in $NH_4^+$ concentration, with mean
concentration of 6.3±2.5 µg m$^{-3}$, can be observed during the period from 17:00 LT on
DOY 114 to 10:00 LT on DOY 120 (Fig. 3b). The mean $N_{ccn}$ during this period varied
from $3.5 \pm 1.0 \times 10^3$ cm$^{-3}$ to $4.0 \pm 1.1 \times 10^3$ cm$^{-3}$ at SS ranging of 0.2% to 1.0%. In contrast,
after DOY 120, the concentration of $NH_4^+$ (0.67±0.70 µg m$^{-3}$) substantially decreased
by almost 90%, during which the mean $N_{ccn}$ at each SS showed statistically significant
decrease of 8% to 15%, implicative of the vital contribution to CCN of secondary
ammonium salt aerosols.

Another feature depicted in Fig. 3b is the $N_{ccn}$ during the low $NH_4^+$ period may even
exceed the maximal value of $N_{ccn}$ during the high $NH_4^+$ period. To elucidate the
underlying mechanism, the $N_{ccn}$, under each SS, was composited and compared
between the days with $NH_4^+$ concentration higher than the upper quartile and the days
in the lower quartile, yielding some interesting findings. At SS=0.2%, the composited
$N_{ccn}$ under high $NH_4^+$ period was higher than that during low $NH_4^+$ period with
statistical significance level of 0.01. There was no significant difference in $N_{ccn}$ between
the two composited periods at SS of 0.4% and 0.6%. However, the composited $N_{ccn}$
(i.e., only selection of the upper quartile) during the high $NH_4^+$ period was significantly
lower than the composited value during the low $NH_4^+$ period with P<0.01, e.g., 5.1 ±
$0.5 \times 10^3$ cm$^{-3}$ versus $5.3 \pm 0.7 \times 10^3$ cm$^{-3}$ at SS=0.8%, $5.2 \pm 0.5 \times 10^3$ cm$^{-3}$ versus 5.7 ±
$0.7 \times 10^3$ cm$^{-3}$ at SS =1.0%. During the low $NH_4^+$ period, the marine atmospheres over
the observational zones may sometimes receive strong continental inputs and/or marine
traffic emissions, leading to the larger $N_{ccn}$. Enhanced formation of ammonium salt



aerosols during the high $NH_4^+$ period likely canceled out or even overwhelmed
continental inputs and/or marine traffic emissions in increasing $N_{ccn}$ at SS=0.2%.

In addition, fresh marine traffic emissions likely yielded a negligible contribution to
$N_{ccn}$ in the marine atmosphere because of a large amount of aged aerosols from various
sources therein. For example, the mean values of $N_{ccn}$ were $3.2 \times 10^3$ $cm^{-3}$ and $4.5 \times 10^3$
$cm^{-3}$ at SS=0.4% and 1.0% at 08:30-11:30 on DOY110, respectively. They were almost
same as $3.2 \times 10^3$ $cm^{-3}$ at SS=0.4% and $3.8 \times 10^3$ $cm^{-3}$ at SS=1.0% before 06:00 on that
day. The mean values of $N_{cn}$, however, largely increased from $6.5 \pm 0.8 \times 10^3$ $cm^{-3}$ before
06:00 to $1.3 \pm 0.3 \times 10^4$ $cm^{-3}$ at 08:30-11:30 when the ship cruised across the Yangtze
River estuary (Fig. 3b).

*3.3 Spatiotemporal variations in CCN activation and Kappa values*
AR values at SS of 0.4% and 1.0% were examined in the section, shown in Fig. 3c. At
SS=0.4%, AR values largely varied from 0.06 to 0.92 with the median value of 0.51.
Specifically, AR values narrowly varied around $0.51 \pm 0.04$ at 00:00-06:00 LT on
DOY110. At 08:00-21:00 LT on that day when the ship cruised across the Yangtze River
estuary, the AR values were substantially decreased to $0.26 \pm 0.06$ concurrently with
approximate 200% increase in $N_{cn}$ values, i.e., $N_{cn}$ value of $6.5 \pm 0.8 \times 10^3$ $cm^{-3}$ at
00:00-06:00 LT and $2.0 \pm 0.7 \times 10^4$ $cm^{-3}$ at 08:00-21:00 LT on DOY110 (Fig. 3a). The
AR values then exhibited an oscillating increase from DOY 111 to DOY113. Low AR
values of $0.12 \pm 0.04$ were suddenly observed at 10:00-18:00 LT on DOY114 in
presence of strong new particle signals transported from the upwind continental
atmosphere, as discussed later. AR values, however, reached $0.34 \pm 0.04$ at 06:00-08:00
LT and $0.39 \pm 0.08$ at 19:00-24:00 LT on DOY114 with the new particle signals largely
reduced. Even excluding the AR values on DOY 114, a significant difference was still
obtained between AR values of $0.61 \pm 0.12$ during the high $NH_4^+$ period and those of
$0.55 \pm 0.17$ during the low $NH_4^+$ period. Enhanced formation of ammonium salts
seemingly increased CCN activity to some extent. At SS=1.0%, AR values showed





large fluctuation with the median value of 0.57± 0.17 (Fig. 3c) and the temporal trend
was similar to that at SS=0.4%.

To minimize the impact from particle sizes, *Kappa* values were further investigated. As
was reported by Phillips et al. (2018), *Kappa* values in a high time resolution usually
exhibited a broad distribution, reflecting the complexity due to various of factors. To
reveal the key factors in determining *Kappa* values in a large spatiotemporal scale, the
daily *Kappa* values of atmospheric aerosols were estimated, on basis of the daily mean
$N_{ccn}$ and the size distributions of particle number concentration from DOY 110-118 (Fig.
3c). Please note that for DOY 110, considering large differences of particle number
concentration between 00:00-06:00 and 08:00-21:00 (Fig. S1), *Kappa* values were
calculated separately for these two periods. At SS=0.4% (green dashed line in Fig. 3c),
the estimated *Kappa* values were as high as 0.66 at 00:00-06:00 LT while it decreased
to 0.37 at 08:00-21:00 LT on DOY110. The *Kappa* value varied narrowly from 0.46 to
0.55 on DOY 111-113, 115 and 117, implying that inorganic aerosols such as
completely and incompletely neutralized ammonium salts may yield a large
contribution to the $N_{ccn}$. These values were generally consistent with reported
observations in most of marine atmospheres. For example, Cai et al. (2017) reported
the *Kappa* value around 0.5 for particles with sizes of 40-200 nm at a marine site in
Okinawa and sulfate to be the dominant component of aerosol particles on 1-9
November 2015, and a similar *Kappa* value in spring 2008 over this site was reported
by Mochida et al. (2010). Royalty et al. (2017) reported *Kappa* values for 48, 96, and
144 nm dry particles to be 0.57 ± 0.12, 0.51 ± 0.09, and 0.52 ± 0.08 in the subtropical
North Pacific Ocean and sulfate-like particles contributing at most 77–88% to the total
aerosol number concentration. *Kappa* values over the Atlantic Ocean were observed
around 0.54 ± 0.03 for 284 nm particles (Phillips et al., 2018).

The estimated *Kappa* values sometimes reached 0.66-0.67 (i.e., on DOY 116), which
may be related to unidentified factors. For example, O'Dowd et al. (2014) proposed that
some organics derived from sea-spray aerosols may also increase the $N_{ccn}$, to some


extent, by reducing surface intension, leading to increase of *Kappa* values. A small
fraction of sea-salt aerosols in submicron particles may also increase *Kappa* values
since its *Kappa* value was as high as 1.3 (O'Dowd et al., 2004; O'Dowd et al., 1997).
The *Kappa* value of 0.29 was obtained on DOY118, close to *Kappa* values widely
observed for continental atmospheric aerosols (~0.3) (Andreae and Rosenfeld, 2008;
Poschl et al., 2009; Rose et al., 2010). The estimated *Kappa* value largely decreased to
0.15 on DOY114 when new particle formation (NPF) occurred, with detailed discussion
in section 3.5. Moreover, at SS of 1.0%, the estimated *Kappa* value was always smaller
than 0.2. The *Kappa* value of organics was commonly assumed as 0.1 (Cai et al., 2017;
Rose et al., 2011; Singla et al., 2017). In general, the fraction of organics in nanometer
particles increases with decreasing particle sizes (Cai et al., 2017; Crippa et al., 2014;
Rose et al., 2011; Rose et al., 2010). A combination of the two factors likely led to
overall *Kappa* values estimated at SS=1.0% to be much lower.

*3.4 Particle number size distributions and CCN activation associated with marine*
*traffic emissions and aerosol aging*
The particle number size distributions during DOY 110-118, shown in Fig. 4, can be in
general classified into two categories. Category 1 occurred on DOY110-114, when
particle number concentrations were mainly distributed at the Aitken mode, whereas
the accumulation mode was generally undetectable. Category 2 occurred on DOY115-
118, when the accumulation mode can be clearly identified and generally dominated
over the Aitken mode. Hoppel W. A. (1986) proposed cloud-modified aerosols to be
mainly distributed at 80-150 nm in the remote tropical Atlantic and Pacific oceans.
Cloud-modified aerosols are quietly common in remote marine atmosphere, likely
leading to the dominate accumulation mode particles to be observed on DOY115-118.
Occasionally, the Aitken mode dominated over the accumulation mode on some day
such as DOY 118. To further dive into the sources of different modes of particles, three-
day of DOY112, DOY 116 and DOY118 were selected.





On DOY 112, the Aitken mode particles accounted for approximately 60% of the total
particle number concentration (Fig. 5a), with median Aitken mode diameters around
54±8 nm. Like the observations over the Yangtze River estuary, the mean value of $N_{cn}$
increased by approximately 50% concurrently with a decrease in the median Aitken
mode diameters by ~9 nm at 05:30 – 11:40 LT against those at the early morning before
05:30 LT (Fig. 5b).    Concomitantly, the AR values decreased to 0.31±0.09 at SS of
0.4%, with similar AR decrease at SS of 1.0%, and the lowest AR and *Kappa* values
occurring at 06:00-07:00 LT at SS of both 0.4% and 1.0%. All these results pointed
towards the increase in Aitken mode particles at 05:30 – 11:40 LT to be likely derived
from enhanced marine traffic contributions carried by the onshore wind from the south
(Fig. S10). During other time on DOY112, the onshore wind may also carry the marine-
traffic derived particles to the observational sea zones. However, the marine-traffic
derived particles likely aged to some extent, e.g., the median Aitken mode diameters
exhibited an oscillating increase from approximately 50 nm at 19:00 to approximately
70 nm at 24:00 LT with the particle growth rate of ~4 nm hour$^{-1}$. The AR values,
however, narrowly varied around 0.47±0.03 at SS=0.4% and 0.52±0.05 at SS=1.0%
during the particle growth period. The *Kappa* values at SS=0.4% gradually decreased
from 0.56 at 19:00 to 0.41 at 23:00 LT, reflecting more aged marine-traffic derived
particles growing into CCN size.

On DOY 116, the accumulation mode particles instead of Aitken mode particles
dominantly contributed to $N_{cn}$ (Fig. 5d), under the marine air influence from the
northeast (Fig. S12). The median accumulation mode diameters narrowly varied around
135±5 nm at 01:00-13:00 LT and 102±5 nm at 16:20-24:00 LT with the transition period
in between (Fig. 5e). The AR and *Kappa* values, however, showed no statistically
significant difference during the two periods at SS of 0.4% and 1.0%, implying that the
size change in accumulation mode particles showed a negligible influence on the CCN
activation. Hourly variations in AR and *Kappa* values may be associated with other
factors, e.g., chemical composition, mixing state, etc. (Gunthe et al., 2011; Rose et al.,

419    2011).




On DOY 118, under the influence of mixture from the marine and coastal areas from
the northeast (Fig. S13), the accumulation mode particles generally dominated the
contribution to $N_{cn}$ while the reverse was true in some occasions (Fig. 5g,h). The median
accumulation mode diameters exhibited an oscillating increase from approximately 100
nm to 130 nm at 00:00-08:00 LT, narrowly varied around 133±5 nm at 08:00-13:00 LT,
and then exhibited an oscillating decrease down to approximately 100 nm at 20:00 LT.
The AR values and *Kappa* values at SS=0.4%, however, exhibited an inverted bell-
shape with the lowest values at 0.31 and 0.11 at 13:00. The decreases in AR values and
*Kappa* may be related to organic condensed on accumulation mode particles since the
median accumulation mode diameters were almost largest at 13:00. The Aitken mode
particles evidently enhanced at 14:00-15:00, but the influence on AR values and *Kappa*
values at SS=0.4% was undetectable (Fig. 5i).

*3.5 The long-range transport of grown new particles on DOY 114*
No hour-long sharp increase in number concentration of nucleation mode particles (<
20 nm) was observed during the period from DOY 110 to DOY 118, except on DOY114
(Fig. 4). According to the conventional definition of NPF events (Dal Maso et al., 2005;
Kulmala et al., 2004), the occurrence frequency of NPF events was low in this study.
Unlike continental atmospheres where a high occurrence frequency of NPF events has
been observed globally in spring (Kerminen et al., 2018; Kulmala et al., 2004), a low
occurrence frequency reportedly occurred over the seas during the "Meiyu (plum-rain)
season" in spring because of frequent rainy, foggy or cloudy weather conditions (Zhu
et al., 2019). Lack of NPF events in the marine atmospheres implied $N_{cn}$ and $N_{ccn}$ to be
mainly contributed by primarily emitted aerosols and their aged products.
During the period of 10:00-18:00 LT on DOY 114, a large increase in number
concentrations of Aitken mode particles (Fig. 6a) likely reflected the long-range
transport of grown new particles from upwind continental atmospheres (Fig. S11). The
size distributions of particle number concentration showed a dominant Aitken mode at



10:00-18:00 LT, when spatiotemporal variations in $N_{cn}$ and median Aitken mode
diameters exhibited bell-shape patterns (Fig. 6b). The median Aitken mode diameters
increased from 26 nm at 10:00 LT to 33 nm at 12:00-13:00 LT and then decreased to 20
nm prior to the signal disappeared, likely reflecting the growth and shrinkage of the
Aitken mode particles (Yao et al., 2010; Zhu et al., 2019). The median Aitken mode
diameters were evidently smaller than the values, i.e., 40-50 nm for Aitken mode
particles, observed over the Yangtze River estuary on DOY 112 (Fig. 5a). Moreover,
the number concentrations of 20-40 nm particles increased by 5.8 times at 12:00-13:00
LT compared to the mean value at 06:00-09:00 LT while the total number
concentrations of particles greater than 90 nm increased by only 67%. These results
implied the largely increased number concentrations of Aitken mode particles with a
dynamic change in mode diameter observed at 10:00-18:00 LT unlikely to be caused
by primarily emitted and aged particles from marine traffic emissions or other
combustion sources. The observations of gaseous and particulate species, during the
same period, implied air masses to be well-aged and less polluted. For instance, the
measured hourly average mixing ratios of $SO_2$ was no larger than 1.2 ppb (Fig. 6c) and
the hourly average concentrations of $NH_4^+$ in $PM_{2.5}$ were smaller than 2 μg m$^{-3}$ (Fig.
3b). In addition, the concentrations of $K^+$ were below 0.3 μg m$^{-3}$, suggesting negligible
contributions from biomass burning (Fig. 6e).

Before 09:00 LT, a much weaker spike of nucleation mode particles was intermittently
observed (Fig. 6a). The weak and intermittent NPF seems to occur in the marine
atmospheres before 09:00 LT when no apparent growth of new particles was observed.
Possibly due to the transport from the continent (Fig. S11) and an increase in the
condensational sink around 10:00 am (Fig. 6a), the weak NPF signal gradually dropped
to a negligible level half an hour later, concomitant with a large increase in the number
concentrations of Aitken mode particles at 10:00-18:00 LT.

$N_{ccn}$ at SS=0.4% increased from $1.2 \times 10^3$ cm$^{-3}$ at 06:00-09:00 LT to the peak value of
$2.3 \times 10^3$ cm$^{-3}$ at 12:00 LT, with increase of 92%, and $N_{ccn}$ at SS=1.0% increased from
$1.6 \times 10^3$ cm$^{-3}$ to $4.0 \times 10^3$ cm$^{-3}$, with increase of 150% (Fig. 6d). The net increase in
$N_{ccn}$ at SS=0.4% likely reflected the contribution from pre-existing particles since new
particles with the diameter less than 50 nm were unlikely activated as CCN at such low
SS (Li et al., 2015; Ma et al., 2016; Wu et al., 2016). The larger net increase in $N_{ccn}$ at
SS=1.0% may reflect the contributions mixed from pre-existing particles and grown
new particles. The high SS can activate particles as CCN with diameters down 40 nm
(Dusek et al., 2006; Li et al., 2015). The invasion of grown new particles also led to the
AR values largely decreased from 0.3 to 0.1 at SS=0.4%, and from 0.4 to 0.2 at SS=1.0%
(Fig. 6e). After 18:00 LT, the AR values retuned to 0.3-0.4 at SS=0.4% and 0.4-0.6 at
SS=1.0%. When the calculated *Kappa* values were examined (Fig. 6c), they decreased
from 0.4 to 0.1-0.2 at SS=0.4%. The value returned to 0.3 at 18:00-19:00 LT (FMPS
was temporarily malfunctioned after 19:20 LT). The *Kappa* values were below 0.2 at
SS=1.0% on that day. The decreases in AR values and *Kappa* values at two SS were
likely caused by organic vapor condensed on preexisting particles and new particles
(Wu et al., 2016; Zhu et al., 2019).

*3.6 Correlations of $N_{cn}$ and $N_{ccn}$ with $SO_2$ in self-ship plumes and ambient air*
When self-ship emission signals were detected, the observational values included a
combination of contributions from self-ship emissions and ambient concentrations.
Although ambient $N_{cn}$ was negligible in comparison with $N_{cn}$ derived from self-ship
emissions, it was not the case for $N_{ccn}$ and $SO_2$. Based on the minutely data, the signal
was considered as vessel-self emission when both $N_{cn}$ greater than 50,000 cm$^{-3}$ and $SO_2$
greater than 5 ppb. The composited data was then used to derive the hourly average $N_{cn}$,
$N_{ccn}$ and $SO_2$, which was then subtracted by the ambient hourly mean value during the
preceding hour with relatively clean conditions (i.e., concentration of $N_{cn}$ lower than
10,000 cm$^{-3}$, $SO_2$ lower than 2.5 ppb). Please note uncertainties exist in terms of the
criteria and separation between self-ship and ambient signals, however, minimal impact
is expected in the relationship examined below.



Fig. 7a showed correlations of $N_{cn}$ and $N_{ccn}$ with mixing ratio of $SO_2$ in self-ship plumes,
prefixed by $\Delta$ for $N_{cn}$, $N_{ccn}$ and $SO_2$ to implicate the removal of ambient signals. . A
good correlation of 0.66 for $R^2$ (P<0.01) was obtained and the slope indicates that $N_{cn}$
increase by $1.4\times10^4$ cm$^{-3}$ for each ppb increase of $SO_2$ resulted from ship emission
(Fig. 7a). High emissions of $N_{cn}$ were generally reported in engine exhausts with high
sulfur-content diesel to be used (Yao et al., 2007; Yao et al., 2005). In regard of $N_{ccn}$ at
SS of 0.2% to 1.0% (Fig. 7b), it increases from 30 cm$^{-3}$ to 170 cm$^{-3}$ per 1 ppb increase
of $SO_2$, showing statistical significant correlation at 99$^{th}$ confidence level. The
contribution ratio of $SO_2$ to $N_{ccn}$ is 0.002 (SS of 0.2%), 0.004(SS of 0.4%) and 0.012
(SS of 1.0%) to that of $N_{cn}$, in general consistent with the previous study by Ramana
and Devi (2016), in which a range of 0.0012–0.57 was observed for CCN at 0.4% in
Bay of Bengal during Aug 13–16, 2012.

The correlations of hourly averaged $N_{cn}$ and $N_{ccn}$ with $SO_2$ in ambient air were examined
and showed in Fig. 7c,d. The data was segmented into pieces based on $SO_2$ with interval
of 0.2 ppb. A good correlation between the averaged $N_{cn}$ and $SO_2$ were obtained with
$R^2$ of 0.80 (P<0.01) and 1 ppb increase in $SO_2$ likely increased $N_{cn}$ by $1.6\times10^3$ cm$^{-3}$
(Fig. 7c). The increase in $N_{cn}$ with $SO_2$ may reflect the contribution from primary
emissions. An intercept was, however, as large as $3.9\times10^3$ cm$^{-3}$, likely representing the
contribution from well-aged aerosols.

Hourly averaged $N_{ccn}$ at different SS generally increased with increase of ambient $SO_2$
(Fig. 7d). A good correlation between the averaged $N_{ccn}$ and $SO_2$ were obtained with
$R^2$=0.78-0.91 (P<0.01). 1 ppb increase in $SO_2$ likely increased $N_{ccn}$ by $0.6\times10^3$ to 0.8
$\times10^3$ cm$^{-3}$ at SS from 0.2% to 1.0%. The increase in $N_{ccn}$ with $SO_2$ may also reflect the
contribution from primary emissions. The intercepts of $2.2\times10^3$-$2.7\times10^3$ cm$^{-3}$ at
different SS were likely contributed by well-aged aerosols. The relationship may be
used as an estimation of $N_{ccn}$ in marine atmospheres over China marginal seas, when
no measurements of CCN were available whereas ambient $SO_2$ can be estimated from



web-based satellite data.

**4. Conclusions**
Spatiotemporal variations in ambient $N_{cn}$ and $N_{ccn}$ were studied during a cruise
campaign on DOY 110-135 over China marginal seas. The mean values of $N_{cn}$ ($8.1\times10^3$
$cm^{-3}$) and $N_{ccn}$ at SS of 0.2%-1.0% ($3.2$ -$3.9 \times10^3$ $cm^{-3}$) were approximately one order
of magnitude larger than those in remote clear marine atmospheres, indicating
overwhelming contributions from non-sea-spray aerosols such as marine traffic
emissions, the long-range continental transport and others.

Observed self-ship emission signals showed fresh marine traffic emissions can be
important sources of $N_{cn}$, but a minor source of $N_{ccn}$ in the marine atmosphere. The
signals showed that 1 ppb increase in $SO_2$ corresponds to $1.4\times10^4$ $cm^{-3}$ increase in $N_{cn}$
and 30-170 $cm^{-3}$ increase in $N_{ccn}$ at SS=0.2-1.0%. Data analysis showed that marine
traffic emissions largely increased $N_{cn}$ over their heavily travelled sea zones in daytime.

In ambient marine air, the growth of marine traffic derived particles led to a decrease
in estimated bulk kappa values at 0.4% possibly because some of these particles
enriched in organics grew into CCN size. However, strong formation of ammonium
salts led to aerosol aging, and significantly increased $N_{ccn}$ at SS of 0.2-1.0% in
comparison with those observed during the period poor in ammonium salt aerosols in
$PM_{2.5}$ with P<0.01. The estimated bulk *Kappa* values from the daily average values
varied from 0.46 to 0.55 at SS=0.4% in most of marine atmospheres, indicating
inorganic ammonium aerosols may dominantly contribute to the $N_{ccn}$ at SS of 0.4%.
The particle number size distributions showed the high bulk *Kappa* values could be
related to cloud-modified aerosols, which likely led to a large extent of degradation of
organics and subsequently lost from the particle phase.

Humid marine ambient air led to NPF events rarely occurring therein. The dominant



onshore winds occurred most of the measurement periods, and should carry primary
aerosols and their aged products rather than secondarily formed aerosols to the
observational zone. During an occasion when offshore winds blew from the northwest
(Fig. S11), new particle signals transported from the continent can be clearly observed.
However, NPF in the marine atmosphere was too weak to be important. The transported
new particles from the continent yielded the maximal increase in $N_{ccn}$ by 92% at SS of
0.4% and 150% at SS of 1.0%. However, consistent with those reported in literature,
the estimated *kappa* values largely decreased from 0.4 to 0.1-0.2 at SS=0.4% during
most time of the continent-transported NPF event because of the *kappa* value of organic
condensation vapor as low as 0.1.

**Competing interests**. The authors declare that they have no conflict of interest.
**Author contributions.** YG and XY designed the research, YG, DZ and XY performed
the analysis, JW and HG helped on the interpretation of the results, and all co-authors
contributed to the writing of the paper.
**Acknowledgment**
This research is supported by the National Key Research and Development Program in
China (grant no. 2016YFC0200504) and the Natural Science Foundation of China
(grant no. 41576118).

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

Contributions from DMS and ship emissions to CCN observed over the summertime
North Pacific, Atmos. Chem. Phys., 10, 1287-1314, DOI 10.5194/acp-10-1287-2010,
2010.
Leng, C., Cheng, T., Chen, J., Zhang, R., Tao, J., Huang, G., Zha, S., Zhang, M., Fang,
697       W., Li, X., and Li, L.: Measurements of surface cloud condensation nuclei and
aerosol activity in downtown Shanghai, Atmos. Environ., 69, 354-361,
10.1016/j.atmosenv.2012.12.021, 2013.
Li, K., Zhu, Y., Gao, H., and Yao, X.: A comparative study of cloud condensation nuclei
measured between non-heating and heating periods at a suburb site of Qingdao in the
North China, Atmos. Environ., 112, 40-53, 10.1016/j.atmosenv.2015.04.024, 2015.
Li, M., Liu, H., Geng, G., Hong, C., Liu, F., Song, Y., Tong, D., Zheng, B., Cui, H., Man,
H., Zhang, Q., and He, K.: Anthropogenic emission inventories in China: a review,
Natl. Sci. Rev., 4, 834-866, 10.1093/nsr/nwx150, 2017.
Lin, Y. C., Chen, J. P., Ho, T. Y., and Tsai, I. C.: Atmospheric iron deposition in the
northwestern Pacific Ocean and its adjacent marginal seas: The importance of coal
burning, Global. Biogeochem. Cy., 29, 138-159, 10.1002/2013GB004795, 2015.
Liu, F., Zhang, Q., A., R. J. v. d., Zheng, B., Tong, D., Yan, L., Zheng, Y., and He, K.:
Recent reduction in NO x emissions over China: synthesis of satellite observations
and emission inventories, Environ. Res. Lett., 11, 114002, 2016.
Ma, N., Zhao, C. S., Tao, J. C., Wu, Z. J., Kecorius, S., Wang, Z. B., Gross, J., Liu, H.
713       J., Bian, Y. X., Kuang, Y., Teich, M., Spindler, G., Muller, K., van Pinxteren, D.,
Herrmann, H., Hu, M., and Wiedensohler, A.: Variation of CCN activity during new


particle formation events in the North China Plain, Atmos. Chem. Phys., 16, 8593-
8607, 10.5194/acp-16-8593-2016, 2016.
Mochida, M., Nishita-Hara, C., Kitamori, Y., Aggarwal, S. G., Kawamura, K., Miura,
K., and Takami, A.: Size-segregated measurements of cloud condensation nucleus
activity and hygroscopic growth for aerosols at Cape Hedo, Japan, in spring 2008, J.
Geophys. Res., 115, 10.1029/2009jd013216, 2010.
Nair, V. S., Nair, J. V., Kompalli, S. K., Gogoi, M. M., and Babu, S. S.: Cloud
Condensation Nuclei properties of South Asian outflow over the northern Indian
Ocean during winter, Atmos. Chem. Phys.Discuss., 10.5194/acp-2019-828, 2019.
O'Dowd, C., Ceburnis, D., Ovadnevaite, J., Vaishya, A., Rinaldi, M., and Facchini, M.
C.: Do anthropogenic, continental or coastal aerosol sources impact on a marine
aerosol signature at Mace Head?, Atmos. Chem. Phys., 14, 10687-10704,
10.5194/acp-14-10687-2014, 2014.
O'Dowd, C. D., Smith, M. H., Consterdine, I. E., and Lowe, J. A.: Marine aerosol, sea-
salt, and the marine sulphur cycle: A short review, Atmos. Environ., 31, 73-80, Doi
10.1016/S1352-2310(96)00106-9, 1997.
O'Dowd, C. D., Facchini, M. C., Cavalli, F., Ceburnis, D., Mircea, M., Decesari, S.,
Fuzzi, S., Yoon, Y. J., and Putaud, J. P.: Biogenically driven organic contribution to
marine aerosol, Nature, 431, 676-680, 10.1038/nature02959, 2004.
Park, M., Yum, S. S., Kim, N., Cha, J. W., Shin, B., and Ryoo, S.-B.: Characterization
of submicron aerosols and CCN over the Yellow Sea measured onboard the Gisang
1 research vessel using the positive matrix factorization analysis method, Atmos.
Res., 214, 430-441, 10.1016/j.atmosres.2018.08.015, 2018.
Petters, M. D., and Kreidenweis, S. M.: A single parameter representation of hygroscopic
growth and cloud condensation nucleus activity, Atmos. Chem. Phys., 7, 1961-1971,
DOI 10.5194/acp-7-1961-2007, 2007.
Phillips, B. N., Royalty, T. M., Dawson, K. W., Reed, R., Petters, M. D., and Meskhidze,
N.: Hygroscopicity- and Size-Resolved Measurements of Submicron Aerosol on the
East Coast of the United States, J. Geophys. Res.-Atmos., 123, 1826-1839,
10.1002/2017JD027702, 2018.



Pöschl, U., Rose, D., & Andreae, M. O. (2009). Climatologies of Cloud-related Aerosols.

Part 2: Particle Hygroscopicity and Cloud Condensation Nucleus Activity. In J.

Heintzenberg, & R. J. Charlson (Eds.), Clouds in the Perturbed Climate System:

Their Relationship to Energy Balance, Atmospheric Dynamics, and Precipitation (pp.

58-72). Cambridge: MIT Press.

Quinn, P. K., and Bates, T. S.: The case against climate regulation via oceanic

phytoplankton sulphur emissions, Nature, 480, 51-56, 10.1038/nature10580, 2011.

Quinn, P. K., Collins, D. B., Grassian, V. H., Prather, K. A., and Bates, T. S.: Chemistry

and Related Properties of Freshly Emitted Sea Spray Aerosol, Chem. Rev., 115,

4383-4399, 10.1021/cr500713g, 2015.

Ramana, M. V., and Devi, A.: CCN concentrations and BC warming influenced by

maritime ship emitted aerosol plumes over southern Bay of Bengal, Sci. Rep., 6,

30416, 10.1038/srep30416, 2016.

Rose, D., Gunthe, S. S., Mikhailov, E., Frank, G. P., Dusek, U., Andreae, M. O., and

Pöschl, U.: Calibration and measurement uncertainties of a continuous-flow cloud

condensation nuclei counter (DMT-CCNC): CCN activation of ammonium sulfate

and sodium chloride aerosol particles in theory and experiment, Atmos. Chem. Phys.,

8, 1153–1179, https://doi.org/10.5194/acp-8-1153-2008, 2008.

Rose, D., Nowak, A., Achtert, P., Wiedensohler, A., Hu, M., Shao, M., Zhang, Y.,

Andreae, M. O., and Pöschl, U.: Cloud condensation nuclei in polluted air and

biomass burning smoke near the mega-city Guangzhou, China - Part 1: Size-resolved

measurements and implications for the modeling of aerosol particle hygroscopicity

and CCN activity, Atmos. Chem. Phys., 10, 3365-3383, DOI 10.5194/acp-10-3365-

2010, 2010.

Rose, D., Gunthe, S. S., Su, H., Garland, R. M., Yang, H., Berghof, M., Cheng, Y. F.,

Wehner, B., Achtert, P., Nowak, A., Wiedensohler, A., Takegawa, N., Kondo, Y., Hu,

771       M., Zhang, Y., Andreae, M. O., and Pöschl, U.: Cloud condensation nuclei in polluted

air and biomass burning smoke near the mega-city Guangzhou, China – Part 2: Size-

resolved aerosol chemical composition, diurnal cycles, and externally mixed weakly

CCN-active soot particles, Atmos. Chem. Phys., 11, 2817-2836, 10.5194/acp-11-



2817-2011, 2011.

Rosenfeld, D., Zhu, Y. N., Wang, M. H., Zheng, Y. T., Goren, T., and Yu, S. C.: Aerosol-

driven droplet concentrations dominate coverage and water of oceanic low-level

clouds, Science, 363, 10.1126/science.aav0566, 2019.

Royalty, T. M., Phillips, B. N., Dawson, K. W., Reed, R., Meskhidze, N., and Petters, M.

D.: Aerosol Properties Observed in the Subtropical North Pacific Boundary Layer, J.

Geophys. Res.-Atmos., 122, 9990-10012, 10.1002/2017JD026897, 2017.

Ruehl, C. R., Chuang, P. Y., and Nenes, A.: Distinct CCN activation kinetics above the

marine boundary layer along the California coast, Geophys. Res. Lett., 36, L15814,

10.1029/2009gl038839, 2009.

Saliba, G., Chen, C. L., Lewis, S., Russell, L. M., Rivellini, L. H., Lee, A. K. Y., Quinn,

P. K., Bates, T. S., Haentjens, N., Boss, E. S., Karp-Boss, L., Baetge, N., Carlson, C.

787        A., and Behrenfeld, M. J.: Factors driving the seasonal and hourly variability of sea-

spray aerosol number in the North Atlantic, Proc. Natl. Acad. Sci. U.S.A., 116,

20309-20314, 10.1073/pnas.1907574116, 2019.

Sato, Y., and Suzuki, K.: How do aerosols affect cloudiness?, Science, 363, 580-581,

10.1126/science.aaw3720, 2019.

Si, Y. D., Yu, C., Zhang, L., Zhu, W. D., Cai, K., Cheng, L. X., Chen, L., and Li, S. S.:

Assessment of satellite-estimated near-surface sulfate and nitrate concentrations and

their precursor emissions over China from 2006 to 2014, Sci. Total Environ., 669,

362-376, 10.1016/j.scitotenv.2019.02.180, 2019.

Singla, V., Mukherjee, S., Safai, P. D., Meena, G. S., Dani, K. K., and Pandithurai, G.:

Role of organic aerosols in CCN activation and closure over a rural background site

in Western Ghats, India, Atmos. Environ., 158, 148-159,

10.1016/j.atmosenv.2017.03.037, 2017.

Song, J. W., Zhao, Y., Zhang, Y. Y., Fu, P. Q., Zheng, L. S., Yuan, Q., Wang, S., Huang,

X. F., Xu, W. H., Cao, Z. X., Gromov, S., and Lai, S. C.: Influence of biomass burning

on atmospheric aerosols over the western South China Sea: Insights from ions,

carbonaceous fractions and stable carbon isotope ratios, Environ. Pollut., 242, 1800-

1809, 10.1016/j.envpol.2018.07.088, 2018.





Ueda, S., Miura, K., Kawata, R., Furutani, H., Uematsu, M., Omori, Y., and Tanimoto,
H.: Number-size distribution of aerosol particles and new particle formation events
in tropical and subtropical Pacific Oceans, Atmos. Environ., 142, 324-339,
10.1016/j.atmosenv.2016.07.055, 2016.
Wang, J., Shen, Y., Li, K., Gao, Y., Gao, H., and Yao, X.: Nucleation-mode particle pool
and large increases in Ncn and Nccn observed over the northwestern Pacific Ocean
in the spring of 2014, Atmos. Chem. Phys., 19, 8845-8861, 10.5194/acp-19-8845-
2019, 2019.
Wang, Z. J., Du, L. B., Li, X. X., Meng, X. Q., Chen, C., Qu, J. L., Wang, X. F., Liu, X.
T., and Kabanov, V. V.: Observations of marine aerosol by a shipborne
multiwavelength lidar over the Yellow Sea of China, Lidar Remote Sensing for
Environmental Monitoring Xiv.International Society for Optics and Photonics, 9262,
10.1117/12.2070297, 2014.
Wu, Z. J., Zheng, J., Shang, D. J., Du, Z. F., Wu, Y. S., Zeng, L. M., Wiedensohler, A.,
and Hu, M.: Particle hygroscopicity and its link to chemical composition in the urban
atmosphere of Beijing, China, during summertime, Atmos. Chem. Phys., 16, 1123-
1138, 10.5194/acp-16-1123-2016, 2016.
Yamashita, K., Murakami, M., Hashimoto, A., and Tajiri, T.: CCN Ability of Asian
Mineral Dust Particles and Their Effects on Cloud Droplet Formation, J. Meteor. Soc.
Japan, 89, 581-587, 10.2151/jmsj.2011-512, 2011.
Yao, X. H., Lau, N. T., Fang, M., and Chan, C. K.: Real-time observation of the
transformation of ultrafine atmospheric particle modes, Aerosol. Sci. Tech., 39, 831-
841, 10.1080/02786820500295248, 2005.
Yao, X. H., Lau, N. T., Chan, C. K., and Fang, M.: Size distributions and condensation
growth of submicron particles in on-road vehicle plumes in Hong Kong, Atmos.
Environ., 41, 3328-3338, 10.1016/j.atmosenv.2006.12.044, 2007.
Yao, X. H., Choi, M. Y., Lau, N. T., Lau, A. P. S., Chan, C. K., and Fang, M.: Growth
and Shrinkage of New Particles in the Atmosphere in Hong Kong, Aerosol. Sci. Tech.,
44, 639-650, Pii 924397031,10.1080/02786826.2010.482576, 2010.
Yu, F., and Luo, G.: Simulation of particle size distribution with a global aerosol model:



contribution of nucleation to aerosol and CCN number concentrations, Atmos. Chem.
Phys., 9, 7691-7710, DOI 10.5194/acp-9-7691-2009, 2009.
Zhu, Y. J., Li, K., Shen, Y. J., Gao, Y., Liu, X. H., Yu, Y., Gao, H. W., and Yao, X. H.:
New particle formation in the marine atmosphere during seven cruise campaigns,
Atmos. Chem. Phys., 19, 89-113, 10.5194/acp-19-89-2019, 2019.
Zimmerman, N., Jeong, C.-H., Wang, J. M., Ramos, M., Wallace, J. S., and Evans, G.
J.: A source-independent empirical correction procedure for the fast mobility and
engine   exhaust   particle   sizers,   Atmos.   Environ.,   100,   178-184,
10.1016/j.atmosenv.2014.10.054, 2015.



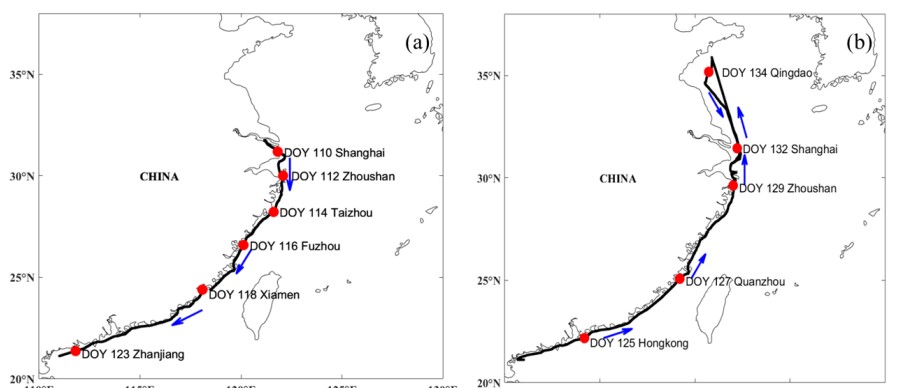

**Fig 1** The ship track during the campaign of 2018, and the blue arrows represented the sailing direction, with southward track (a) and northward track (b).

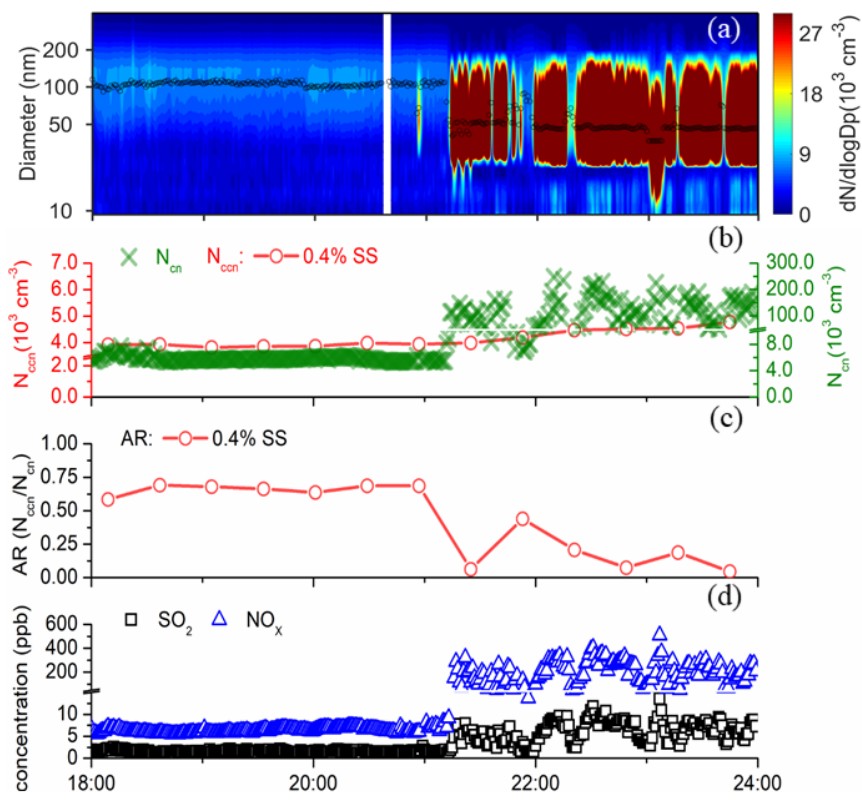

**Fig 2** Contour plot of particle number size distribution with the median mobility mode diameter shown in black hollow circles (a), time series of minutely $N_{cn}$ and half-hourly $N_{ccn}$ at SS=0.4% (b), half-hourly AR values at SS=0.4% (c), $SO_2$ and $NO_x$ at nighttime on DOY 115.

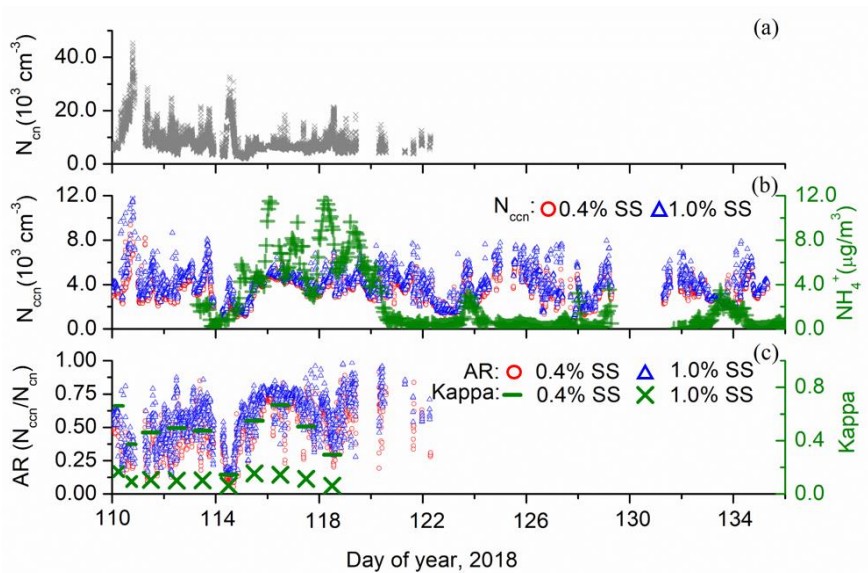

**Fig 3** Time series of minutely $N_{cn}$ from DOY 110 to 122 (a), minutely $N_{ccn}$ at SS of 0.4% and 1.0% during DOY 110-135 and hourly $NH_4^+$ during DOY 113-135 (b), and minutely AR at SS of 0.4% and 1.0% during DOY 110-122 and daily *Kappa* values at SS of 0.4% and 1.0% from DOY 110 to 118 due to data availability (c). Please note that for Fig. 3c, most *Kappa* values were based on daily scale, except on DOY 110, during which two *Kappa* values were calculated from 00:00-06:00 and 08:00-21:00, respectively.





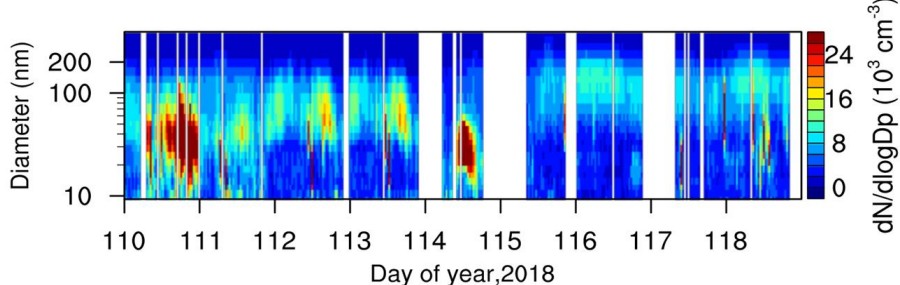

**Fig 4** Contour plot of particle number size distribution on DOY 110-118 with self-ship emission signals removed.

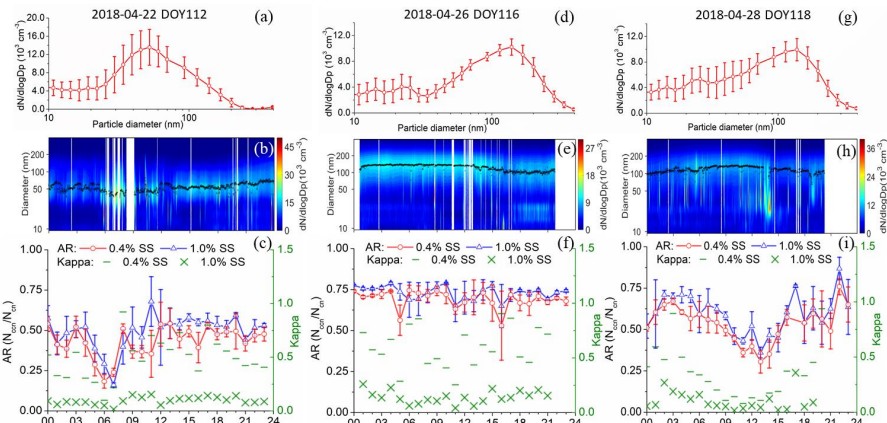

Fig 5 Daily average (top row) and contour plot (middle row) of particle number size distributions, and time series of hourly averaged AR at SS of 0.4% and 1.0% and *Kappa* value on DOY112, DOY116 and DOY118. The bars represent the standard deviation with mean indicated by the hollow circles.



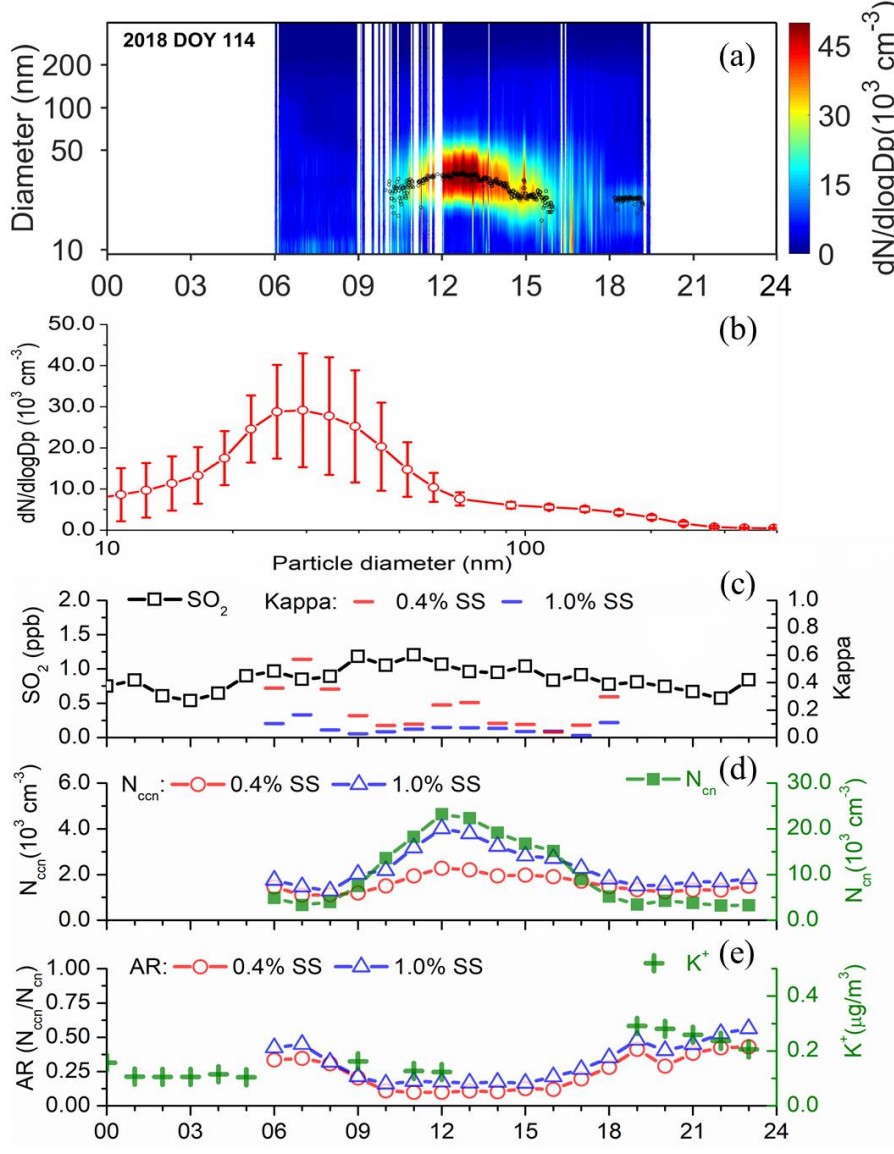

**Fig 6** Contour plot of particle number size distributions for the day of DOY 114 2018 (a), the size distributions of particle number concentration during 10:00 -18:00 LT DOY 114 2018 (b), time series of hourly averaged $SO_2$ and *Kappa* values at SS of 0.4% and 1.0% (c), $N_{ccn}$ at SS of 0.4% and 1.0% (d), and AR values at SS of 0.4% and 1.0% and $K^+$ (e) for the day of DOY 114 2018.

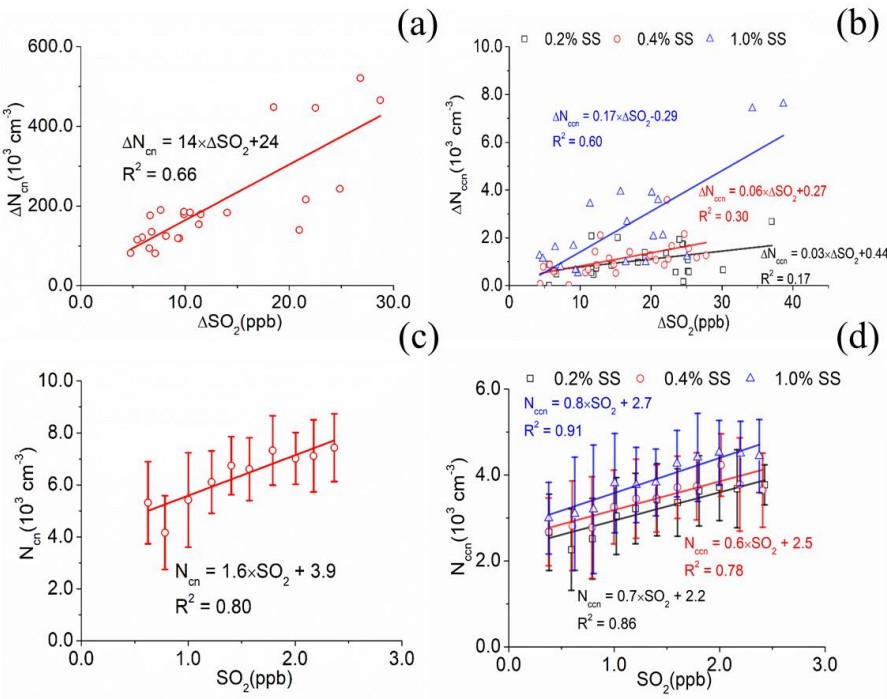

**Fig 7** Correlations of hourly averaged $N_{cn}$ and $N_{ccn}$ with $SO_2$ at SS of 0.2%, 0.4% and 1.0%. For Fig. 7a,b, $\triangle N_{cn}$, $\triangle N_{ccn}$ and $\triangle SO_2$ reflects the impact from self-ship emission after the removal of ambient concentration. For Fig. 7c,d each bar indicates standard deviation with mean value marked as the hollow circles (or triangles, squares), and the interval of $SO_2$ is 0.2 ppb for each bar.



**Table 1.** $N_{cn}$ and $N_{ccn}$, AR and $SO_2$ mixing ratios on DOY 110-135, 2018 over China marginal seas. Please note that Ncn and AR are from 110-122, 2018.

| Variables | Supersaturation (SS) | Ranges | Mean ± standard deviation |
|---|---|---|---|
| $N_{cn}$ ($\times10^3$ cm$^{-3}$) | | 2.0-45 | 8.1±4.4 |
| | SS=0.2% | 0.4-8.8 | 3.2±1.1 |
| | SS=0.4% | 0.5-9.4 | 3.4±1.1 |
| $N_{ccn}$ ($\times10^3$ cm$^{-3}$) | SS=0.6% | 0.5-8.6 | 3.6±1.2 |
| | SS=0.8% | 0.5-11 | 3.8±1.2 |
| | SS=1.0% | 0.6-12 | 3.9±1.4 |
| | SS=0.2% | 0.06-0.89 | 0.49±0.17 |
| | SS=0.4% | 0.06-0.92 | 0.51±0.17 |
| AR | SS=0.6% | 0.10-0.94 | 0.54±0.17 |
| | SS=0.8% | 0.08-0.95 | 0.56±0.17 |
| | SS=1.0% | 0.11-0.98 | 0.57±0.17 |
| $SO_2$ (ppb) | | 0.25-9.7 | 1.7±1.1 |