# Peer review of "Variations in $N_{cn}$ and $N_{ccn}$ over China marginal seas related to marine traffic emissions, new particle formation and aerosol aging"

_Atmospheric Chemistry and Physics, 2020_

## Referee Comment (RC1) · Anonymous Referee #3 · 14 Apr 2020

This study investigated the spatial and temporal variations of Ncn, Nccn and a few other pollutants over China marginal seas. The authors also examined the relationship between Ncn, Nccn and $SO_2$, which is useful to predict Ncn and Nccn when the observation of $SO_2$ is available. This study is interesting and important to understand the characteristics of Ncn and Nccn over the marine atmosphere.

A few specific comments shown below need to be addressed.

1. Line 45-46 (abstract): Moreover, the influences of the transported new particles from the continent on Ncn and Nccn in the marine atmosphere were also investigated.

The sentence illustrated the transport of new particles. Are there any findings or conclusions from this analysis?

2. Line 76-78: different marine atmospheres, e.g., over Mediterranean, Sea of Japan, Bay of Bengal, coast of California and the Northwest Pacific Ocean etc. (Bougiatioti et al., 2009; Ramana and Devi, 2016; Ruehl et al., 2009; Wang et al., 2019; Yamashita et al., 2011).
Are the locations correspond to the five references? It is better to list the reference following each location.

3. Line 115-116
The correlation equations are valuable for a rough estimation of Ncn and Nccn from $SO_2$ when their direct observations are not available.
Please rephrase this sentence. For example, "their direct observation" mean the observation of Ncn and Nccn?

4. Line 153: ECMWF
There is a couple of data sets from ECMWF. What did the authors use, ERA-40, ERA-Interim, or ERA5? Please be specific.

5. Line 157-159: Meanwhile, the data of fire spots was available at the Fire Information for Resource Management System (FIRMS;http://firefly.geog.umd.edu/firemap).
It does not seem the fire spots were used in this study. If not, the descriptions need to be removed.

6. Line 430-431: The Aitken mode particles evidently enhanced at 14:00-15:00
The enhancement here means the number concentration? Please clarify.

Technical corrections:

Line 102. continent al aerosols. An extra space before al, please remove it.

Line 126-128: The FMPS were used; CPC were
The word "were" changed to "was"

Line 150-151: Ambient Ion Monitor-Ion chromatography (AIM-IC)
AIM-IC has been defined at Line 125. Please avoid the duplication of definition.

Line 164: one extra comma. Please delete it. Similar for Line 509.

Line 189: "Fib" should be changed to "Fig"

Line 198: "." needs to be changed to ","

Line 273: "relative" should be revised to "relatively"

Line 382: Hoppel W. A. (1986) proposed cloud-modified aerosols to be …
The citation is not appropriate. Hoppel et al. (1986)

Line 384: Cloud-modified aerosols are quietly common

Line 452. prior to the signal disappeared
Grammar error exist here.

Line 484. with diameters down 40 nm
Changed to "with diameters down to 40 nm"

Nccn at SS of 0.2%-1.0% (3.2 -3.9 $\times 10^3$ cm$^{-3}$)
The values inside the parenthesis should be moved next to Nccn.

---

## Referee Comment (RC2) · Anonymous Referee #2 · 26 May 2020

*Summary:*

This work demonstrates the number concentrations of CN and CCN during a cruise campaign. The manuscript fits well to the scope of ACP. However, I am worried about the method used for retrieving hygrscopicity $\kappa$. This paper is worth to be published, but not in its current form. Thus I recommend it to be resubmitted after the following major comments listed below have been adequately addressed.

*Comments:*

1. Page 5, line 128: If the authors used CPC 3775, then the default $D_{50}$ should be 4 nm. Did you alter the supersaturation to get smaller cut-off size?
2. Page 5, lines 134-139: Please provide the thermodynamic parameterizations for CCNC calibration. What do you mean "collection of CCN"?
3. Page 6, lines 161-171: I don't understand the method to calculate the kappa value, of which critical dry diameter and Sc are needed. I guess there is no measurement of size distribution of CCN (DMA-CCNC), so how could you get the dry diameter?
4. Page 12, line 358-373: there are many assumptions in this section. Could AIM-IC measurement provide some evidence?
5. How you use Hysplit model and fire spots, I did not see it in the main text.
6. There are several grammar mistakes in the text and figures, the language, symbols and labels should be checked carefully.

---

## Author Comment (AC1) · 16 Jun 2020

1. Line 45-46 (abstract): Moreover, the influences of the transported new particles from the continent on Ncn and Nccn in the marine atmosphere were also investigated. The sentence illustrated the transport of new particles. Are there any findings or conclusions from this analysis?

Response: The long-range transport of new particles from the upwind continental atmosphere has been discussed in Section 3.5. A few important findings and conclusions are highlighted below:

[Figure]

"Before 09:00 LT on DOY114 of 2018, a much weaker spike of nucleation mode particles was intermittently observed (Fig. 6a). The weak and intermittent NPF seems to occur in the marine atmospheres before 09:00 LT when no apparent growth of new particles was observed. Possibly due to the transport from the continent (Fig. S12) and an increase in the condensational sink around 10:00 am (Fig. 6a), the weak NPF signal gradually dropped to a negligible level half an hour later, concomitant with a large increase in the number concentrations of Aitken mode particles at 10:00-18:00 LT." This suggested the humid marine atmosphere did not favor regular banana-shape NPF events. However, the NPF event can occur in the upwind continental atmosphere and the grown new particles can be transported and detected in the marine atmosphere under the offshore wind.

2. Line 76-78: different marine atmospheres, e.g., over Mediterranean, Sea of Japan, Bay of Bengal, coast of California and the Northwest Pacific Ocean etc. (Bougiatioti et al., 2009; Ramana and Devi, 2016; Ruehl et al., 2009; Wang et al., 2019; Yamashita et al., 2011). Are the locations corresponding to the five references? It is better to list the reference following each location.

Response: Right, they are corresponding to each other, and the reference has been listed adjacent to the respective site.

3. Line 115-116: The correlation equations are valuable for a rough estimation of Ncn and Nccn from SO2 when their direct observations are not available. Please rephrase this sentence. For example, "their direct observation" mean the observation of Ncn and Nccn?

Response: It has been rephrased.

4. Line 153: ECMWF There is a couple of data sets from ECMWF. What did the authors use, ERA-40, ERA-Interim, or ERA5? Please be specific.

Response: The data used is NCEP GDAS, and this was corrected in the revised

manuscript.

5. Line 157-159: Meanwhile, the data of fire spots was available at the Fire Information for Resource Management System (FIRMS;http://firefly.geog.umd.edu/firemap). It does not seem the fire spots were used in this study. If not, the descriptions need to be removed.

Response: It has been removed. 6. Line 430-431: The Aitken mode particles evidently enhanced at 14:00-15:00 The enhancement here means the number concentration? Please clarify.

Response: Right, it has been revised.

Technical corrections: Line 102. continent al aerosols. An extra space before al, please remove it. Line 126-128: The FMPS were used; CPC were The word "were" changed to "was" Line 150-151: Ambient Ion Monitor-Ion chromatography (AIM-IC) AIM-IC has been defined at Line 125. Please avoid the duplication of definition. Line 164: one extra comma. Please delete it. Similar for Line 509. Line 189: "Fib" should be changed to "Fig" Line 198: "." needs to be changed to "," Line 273: "relative" should be revised to "relatively" Line 382: Hoppel W. A. (1986) proposed cloud-modified aerosols to be . . . The citation is not appropriate. Hoppel et al. (1986) Line 384: Cloud-modified aerosols are quietly common Line 452. prior to the signal disappeared Grammar error exist here. Line 484. with diameters down 40 nm. Changed to "with diameters down to 40 nm" Nccn at SS of 0.2%-1.0% (3.2 -3.9 $\times 10^3$ cm-3) The values inside the parenthesis should be moved next to Nccn.

Response: Sorry for the typos. Agree and revised.

---

## Author Comment (AC2) · 16 Jun 2020

1. Page 5, line 128: If the authors used CPC 3775, then the default D50 should be 4 nm. Did you alter the supersaturation to get smaller cut-off size?

Response: The D50 is 4 nm, and this was revised.

2. Page 5, lines 134-139: Please provide the thermodynamic parameterizations for CCNC calibration. What do you mean "collection of CCN"?

Response: The calibration curve has been added in revised Supporting Information (Fig S1). "collection of CCN" has been changed to "measurement of Nccn" to avoid

confusion.

3. Page 6, lines 161-171: I don't understand the method to calculate the kappa value, of which critical dry diameter and Sc are needed. I guess there is no measurement of size distribution of CCN (DMA-CCNC), so how could you get the dry diameter?

Response: This has been clarified in the revision, i.e., "The Dd was not measured directly and assumed to be equal to the critical diameter for CCN activation (Dcrit). Dcrit was defined as the particle diameter down to which by integrating from the largest diameter with the number concentration equals to CCN concentration (Hung et al., 2014; Cheung et al., 2020)."

4. Page 12, line 358-373: there are many assumptions in this section. Could AIM-IC measurement provide some evidence?

Response: We agree that the weakness indeed exists because of lack of direct measurements of chemical composition of nanometer particles. In the revision, we added "However, the direct measurements of chemical composition of nanometer particles needed to confirm the argument."

It is a common challenge to measure high time-resolution chemical composition of atmospheric nanometer particles. Based on our experience using AMS-measurements in Hong Kong, mass concentrations of chemical species in atmospheric nanometer particles are always negative (Environ. Sci. Technol. 2015, 49, 12, 7170–7178). AIM-IC only measures ionic species in PM2.5, but cannot measure most of organics and black carbon. We are sorry for this because the AIM-IC data cannot allow us saying more on this issue.

5. How you use Hysplit model and fire spots, I did not see it in the main text.

Response: The Hysplit model results are shown in Fig. S11-S14 in the supporting information. There was no fire data used, and the relevant descriptions have been deleted.

6. There are several grammar mistakes in the text and figures, the language, symbols and labels should be checked carefully.

Response: Thanks. We have checked the entire manuscript.
* * *

---

## Author Response (AR2)

Comments to the Author:
The authors have responded most of the referee comments. However, when looking at the revised paper, there are still a few issues that should be considered before accepting this paper for publication. They are listed below.

We thank the editor for carefully reviewing the manuscript, and all the comments have been addressed.

In the introduction the authors discuss marine aerosols in very general terms, but then mention emission trends that are not true in general in the global atmosphere (lines 97-100). Please either mention explicitly here that you refer to Asian emissions, or revise the text according to what has taken place in different continents (In Europe and NA, for example, SO2 emissions have decreased for several decades and also NOx emissions have evolved very differently from those in Asia).

Response: The emission trend is mainly applied to China, and this has been added in the revised manuscript.

line 161: Surface tension is a composition-dependent variable, not a constant. It is usually set constant when applying the equation to calculate the hygroscopicity parameter cappa, but that is a different thing. Please correct the text.

Response: This has been revised as follows:

$\sigma_{s/a}$ represents the surface tension over the interface of the solution and air with the value of 0.072 J m$^{-2}$ applied in this study

Lines 362-369: The minimum diameter of particles at SS of 0.2-1 % is somewhere in the range 50-100 nm, so statement about nanometer-size particles on lines 366-367 is very confusing. Why to talk about nanometer-size (sub-10 nm) particles here? Is this statement even correct? There is often more organics in sub-100 nm particle compared with accumulation mode particles, but the situation in the sub-10 nm size range may be totally different. Please be more careful here what size ranges you are really referring to here.

Response: This has been revised as follows:

In general, the fraction of organics in the nanometer particles increases with decreasing particle size from ~100 nm to ~50 nm

Section 3.6, end of Introduction and Figure 7: I would recommend using the terms "relationship" or "regression equation" rather than "correlation" here. The term "correlation equation" does not sound correct at all.

Response: This has been revised based on the reviewer's suggestion.

The text still contains many grammatical problems (lack of articles, wrong prepositions etc.). Please check out the text once more with the help of a native English speaker.

Response: We have asked a native speaker to check the grammar issues.

[revised manuscript text omitted]